# Generalized Solutions of the Third-Order Cauchy-Euler Equation in the Space of Right-Sided Distributions via Laplace Transform

**Seksan Jhanthanam [1], Kamsing Nonlaopon [1,\*] and Somsak Orankitjaroen [2]**

[1] Department of Mathematics, Khon Kaen University, Khon Kaen 40002, Thailand; Jhanthanam.seksan@gmail.com
[2] Department of Mathematics, Faculty of Science, Mahidol University, Bangkok 10400, Thailand; somsak.ora@mahidol.ac.th
\* Correspondence: nkamsi@kku.ac.th; Tel.: +668-6642-1582

**Abstract:** Using the Laplace transform technique, we investigate the generalized solutions of the third-order Cauchy-Euler equation of the form

$$t^3 y'''(t) + at^2 y''(t) + by'(t) + cy(t) = 0,$$

where $a, b$, and $c \in \mathbb{Z}$ and $t \in \mathbb{R}$. We find that the types of solutions in the space of right-sided distributions, either distributional solutions or weak solutions, depend on the values of $a$, $b$, and $c$. At the end of the paper, we give some examples showing the types of solutions. Our work improves the result of Kananthai (Distribution solutions of the third order Euler equation. *Southeast Asian Bull. Math.* **1999**, *23*, 627–631).

**Keywords:** Cauchy-Euler equation; Dirac delta function; distributional solutions; Laplace transform; weak solutions

## 1. Introduction

Differential equations are used to construct models of reality. Sometimes, the reality we are modeling suggests that some solutions of the differential equations with singularity in the coefficients need not be differentiable in the classical sense. This is where the concept of distributions or generalized functions comes from, which contains the continuous function as a subset. Kanwal [1] was the first who classified the types of solution of the linear differential equation of order $n$ of the form:

$$\sum_{m=0}^{n} a_m(t) \frac{d^m y}{dt^m} = f(t), \tag{1}$$

where $a_m(t) \in C^\infty(\mathbb{R})$ with $a_n(t) \neq 0$ and $f(t)$ is a given distribution. Solution types of the differential Equation (1) can be classified into 3 groups. The first one is the classical solution; it is the one with $n$-order continuously differentiable and satisfies (1) in the ordinary sense. The second one is the weak solution; it is the one with less smoothness but still locally integrable, and satisfies (1) in the distributional sense. The third one is the distributional solution; it is in fact a singular distribution and satisfies (1) in the distributional sense. All these solutions are called *generalized solutions*.

It is well known that the linear homogeneous ordinary differential equations with infinitely-smooth coefficients have no generalized solutions in the sense of the distribution; while the ordinary differential equations with polynomial coefficients such as the Cauchy-Euler equation defined

by Equation (1), where $a_m(t) = c_m t^m$ with $c_m \in \mathbb{R}, c_n = 1$ and $f(t) = 0$, may have a classical solution or a generalized solution in the sense of the distribution; see [2–4] for more details. The method for solving the classical solution of the Cauchy-Euler equation was explained in [5–9].

The generalized solutions in the sense of the distribution of ordinary differential equations can be derived from the theory of distributions. Research in these areas, still developing continually, has opened up many aspects and properties in the theory of differential and functional differential equations.

In case of the weak solutions of certain differential equations, many of them have already been studied. For example, Kananthai and Nonlaopon [10] studied the weak solutions of the compound ultra-hyperbolic equation while Sarikaya and Yildirim [11] studied those of the compound Bessel ultra-hyperbolic equation. More details related to the weak solutions of certain differential equations in the field of theory of distributions are referred to [12–14].

Regarding the distributional solutions, specifically as a series of Dirac delta function and its derivatives, they have been used in several areas of applied mathematics such as the theory of partial differential equations, operational calculus, and functional analysis; in Physics such as quantum electrodynamics. We refer the readers to the papers [15–20] for more details.

For the generalized solutions, Nonlaopon et al. [21] used the Laplace transform technique to study those satisfying the differential equation

$$ty^{(n)}(t) + my^{(n-1)}(t) + ty(t) = 0,$$

where $m, n \in \mathbb{Z}$ with $n \geq 2$ and $t \in \mathbb{R}$. Opio et al. [22] studied those of the differential equation with polynomial coefficients of the form

$$ty^{(n)}(t) + (m-t)y^{(n-1)}(t) - py(t) = 0,$$

where $n, p, m \in \mathbb{N}, n \geq 2$, and $t \in \mathbb{R}$.

Now, we consider the third-order Cauchy-Euler equation of the form

$$t^3 y'''(t) + at^2 y''(t) + bt y'(t) + cy(t) = 0, \tag{2}$$

where $a, b$, and $c \in \mathbb{Z}$ and $t \in \mathbb{R}$. The classical solutions of the above equation are in the space $C(\mathbb{R})$ of continuous functions of the form $t^\lambda$, where $\lambda$ is a real or a complex number; see [8] for more details. The classical solutions of this equation using the Laplace transform technique were also studied by Kim [23]. Moreover, Kananthai [24] studied the generalized solutions of (2), where $a, b = 1$, $c$ is some integer, and $t \in \mathbb{R}$ using the Laplace transform technique. He found that one distributional solution and one weak solution depend on the values of $m$. Next, Sacorn et al. [25] studied the generalized solutions of (2), where $a, b, c$ are real constants and $t \in \mathbb{R}$ using the Laplace transform technique. They found that one distributional solution and one weak solution of (2) depend on the values of $a, b$, and $c$. For this work our goal is to investigate the generalized solutions in the space of right-sided distributions of (2) for the case of one distributional solution, two distributional solutions, three distributional solutions, containing both the distributional solution and weak solution, the linear combination of two distributional solutions and one weak solution, and the linear combination of one distributional solution and two weak solutions.

This paper is organized as follows. Next section we discuss a fundamental idea of Laplace transform. We proceed to Section 3 with the use of Laplace transform technique to our equation with a further investigation. Many examples are then presented to support our main results. Finally, we summarize our work in Section 4.

## 2. Preliminaries

Before we proceed to our main results, the following definitions and concepts are required.

**Definition 1.** *Let $\mathcal{D}$ be the space consisting of all real-valued functions $\varphi(t)$ with continuous derivatives of all orders and compact support. The support of $\varphi(t)$ is the closure of the set of all elements $t \in \mathbb{R}$ such that $\varphi(t) \neq 0$. Then, $\varphi(t)$ is called a test function.*

**Definition 2.** *A distribution $T$ is a continuous linear functional on the space $\mathcal{D}$ of the real-valued functions with infinitely-differentiable and bounded support. The space of all such distributions is denoted by $\mathcal{D}'$.*

For every $T \in \mathcal{D}'$ and $\varphi(t) \in \mathcal{D}$, the value that $T$ has on $\varphi(t)$ is denoted by $\langle T, \varphi(t) \rangle$. Note that $\langle T, \varphi(t) \rangle \in \mathbb{R}$.

**Example 1.**

(i)   *The locally-integrable function $f(t)$ is a distribution generated by the locally-integrable function $f(t)$. Then, we define $\langle f(t), \varphi(t) \rangle = \int_{\Omega} f(t)\varphi(t)dt$, where $\Omega$ is the support of $\varphi(t)$ and $\varphi(t) \in \mathcal{D}$.*
(ii)  *The Dirac delta function is a distribution defined by $\langle \delta(t), \varphi(t) \rangle = \varphi(0)$, and the support of $\delta(t)$ is $\{0\}$.*

A distribution $T$ generated by a locally-integrable function is called a regular distribution; otherwise, it is called a singular distribution.

**Definition 3.** *The kth-order derivative of a distribution $T$, denoted by $T^{(k)}$, is defined by $\left\langle T^{(k)}, \varphi(t) \right\rangle = (-1)^k \left\langle T, \varphi^{(k)}(t) \right\rangle$ for all $\varphi(t) \in \mathcal{D}$.*

**Example 2.**

(i)   $\langle \delta'(t), \varphi(t) \rangle = - \langle \delta(t), \varphi'(t) \rangle = -\varphi'(0)$;
(ii)  $\left\langle \delta^{(k)}(t), \varphi(t) \right\rangle = (-1)^k \left\langle \delta(t), \varphi^{(k)}(t) \right\rangle = (-1)^k \varphi^{(k)}(0)$.

**Definition 4.** *Let $\alpha(t)$ be an infinitely-differentiable function. We define the product of $\alpha(t)$ with any distribution $T$ in $\mathcal{D}'$ by $\langle \alpha(t)T, \varphi(t) \rangle = \langle T, \alpha(t)\varphi(t) \rangle$ for all $\varphi(t) \in \mathcal{D}$.*

**Definition 5.** *Let $M \in \mathbb{R}$ and $f(t)$ be a locally-integrable function satisfying the following conditions:*

(i)   *$f(t) = 0$ for all $t < M$;*
(ii)  *There exists a real number $c$ such that $e^{-ct}f(t)$ is absolutely integrable over $\mathbb{R}$.*

*The Laplace transform of $f(t)$ is defined by:*

$$F(s) = \mathcal{L}\{f(t)\} = \int_M^{\infty} f(t)e^{-st}dt, \tag{3}$$

*where $s$ is a complex variable.*

It is well known that if $f(t)$ is continuous, then $F(s)$ is an analytic function on the half-plane $\Re(s) > \sigma_a$, where $\sigma_a$ is an abscissa of absolute convergence for $\mathcal{L}\{f(t)\}$.

**Definition 6.** *Let $f(t)$ be a function satisfying the same conditions as in Definition 5 and $\mathcal{L}\{f(t)\} = F(s)$. The inverse Laplace transform of $F(s)$ is defined by:*

$$f(t) = \mathcal{L}^{-1}\{F(s)\} = \frac{1}{2\pi i} \lim_{\omega \to \infty} \int_{c-i\omega}^{c+i\omega} F(s)e^{st}ds, \tag{4}$$

*where $\Re(s) > \sigma_a$.*

Recall that the Laplace transform $G(s)$ of a locally-integrable function $g(t)$ satisfying the conditions of Definition 5, that is,

$$G(s) = \mathcal{L}\{g(t)\} = \int_M^\infty g(t)e^{-st}dt, \tag{5}$$

where $\Re(s) > \sigma_a$, can be written in the form $G(s) = \langle g(t), e^{-st} \rangle$.

**Definition 7.** *Let S be the space of test functions of rapid decay containing the complex-valued functions $\phi(t)$ having the following properties:*

(i)  *$\phi(t)$ is infinitely differentiable, i.e., $\phi(t) \in C^\infty(\mathbb{R})$;*

(ii) *$\phi(t)$, as well as its derivatives of all orders vanish at infinity faster than the reciprocal of any polynomial, which is expressed by the inequality:*

$$|t^p\phi^{(k)}(t)| < C_{pk},$$

*where $C_{pk}$ is a constant depending on $p, k$, and $\phi(t)$. Then, $\phi(t)$ is called a test function in the space S.*

**Definition 8.** *A distribution of slow growth or tempered distribution T is a continuous linear functional over the space S of the test function of rapid decay containing the complex-valued functions, i.e., a complex number $\langle T, \phi(t) \rangle$ assigned with the properties:*

(i)  *$\langle T, c_1\phi_1(t) + c_2\phi_2(t) \rangle = c_1 \langle T, \phi_1(t) \rangle + c_2 \langle T, \phi_2(t) \rangle$ for $\phi_1(t), \phi_2(t) \in S$ and constants $c_1, c_2$;*

(ii) *$\lim_{m\to\infty} \langle T, \phi_m(t) \rangle = 0$ for every null sequence $\{\phi_m(t)\} \in S$.*

*We shall let $S'$ denote the set of all distributions of slow growth.*

**Definition 9.** *Let $f(t)$ be a distribution satisfying the following properties:*

(i)  *$f(t)$ is a right-sided distribution, that is $f(t) \in \mathcal{D}'_R$.*

(ii) *There exists a real number c such that $e^{-ct}f(t)$ is a tempered distribution.*

*The Laplace transform of a right-sided distribution $f(t)$ satisfying (ii) is defined by:*

$$F(s) = \mathcal{L}\{f(t)\} = \left\langle e^{-ct}f(t), X(t)e^{-(s-c)t} \right\rangle, \tag{6}$$

*where $X(t)$ is an infinitely-differentiable function with support bounded on the left, which equals one over a neighborhood of the support of $f(t)$.*

For $\Re(s) > c$, the function $X(t)e^{-(s-c)t}$ is a testing function in the space $S$ and $e^{-ct}f(t)$ is in the space $S'$. Equation (6) can be reduced to:

$$F(s) = \mathcal{L}\{f(t)\} = \langle f(t), e^{-st} \rangle. \tag{7}$$

Now, $F(s)$ is a function of $s$ defined over the right half-plane $\Re(s) > c$. Zemanian [26] proved that $F(s)$ is an analytic function in the region of convergence $\Re(s) > \sigma_1$, where $\sigma_1$ is the abscissa of convergence and $e^{-ct}f(t) \in S'$ for some real number $c > \sigma_1$.

**Example 3.** *Let $\delta(t)$ be the Dirac delta function, $H(t)$ be the Heaviside function, and $f(t)$ be a Laplace-transformable distribution in $\mathcal{D}'_R$. If k is a positive integer, then the following hold:*

(i)   *$\mathcal{L}\{(t^{k-1}H(t))/(k-1)!\} = 1/s^k, \quad \Re(s) > 0;$*

(ii)  *$\mathcal{L}\{\delta(t)\} = 1, \quad -\infty < \Re(s) < \infty;$*

(iii) *$\mathcal{L}\left\{\delta^{(k)}(t)\right\} = s^k, \quad -\infty < \Re(s) < \infty;$*

(iv)  *$\mathcal{L}\left\{t^k f(t)\right\} = (-1)^k F^{(k)}(s), \quad \Re(s) > \sigma_1;$*

(v) $\quad \mathcal{L}\left\{f^{(k)}(t)\right\} = s^k F(s), \quad \Re(s) > \sigma_1.$

**Lemma 1.** *If the equation:*

$$\sum_{i=0}^{n} a_i(t) t^i y^{(i)}(t) = 0 \tag{8}$$

*with infinitely-differentiable coefficients $a_i(t)$ and $a_n(0) \neq 0$ has a solution:*

$$y(t) = \sum_{i=0}^{p} a_i \delta^{(i)}(t), \quad a_p \neq 0, \tag{9}$$

*of order p, then:*

$$\sum_{i=0}^{n} (-1)^i a_i(0)(p+i)! = 0. \tag{10}$$

*Conversely, if p is the smallest nonnegative integer root of* (10), *then there exists a pth-order solution of* (9) *at $t = 0$.*

The proof of this Lemma is given in [18].

**Lemma 2.** *Let $\psi(t)$ be an infinitely-differentiable function. Then,*

$$\psi(t)\delta^{(m)}(t) = (-1)^m \psi^{(m)}(0)\delta(t) + (-1)^{m-1} m \psi^{(m-1)}(0)\delta'(t)$$
$$+ (-1)^{m-2} \frac{m(m-1)}{2!} \psi^{(m-1)}(0)\delta''(t) + \cdots + \psi(0)\delta^{(m)}(t), \tag{11}$$

*and:*

$$[\psi(t)H(t)]^{(m)} = \psi^{(m)}(t)H(t) + \psi^{(m-1)}(0)\delta(t) + \psi^{(m-2)}(0)\delta'(t)$$
$$+ \cdots + \psi(0)\delta^{(m-1)}(t). \tag{12}$$

The proof of Lemma 2 is given in [1].

A useful formula that follows from (11), for any monomial $\psi(t) = t^n$, is that:

$$t^n \delta^{(m)}(t) = \begin{cases} 0 & \text{if } m < n, \\ (-1)^n \frac{m!}{(m-n)!} \delta^{(m-n)}(t) & \text{if } m \geq n. \end{cases} \tag{13}$$

## 3. Main Results

In this section, we will state our main results and give their proofs.

**Theorem 1.** *Consider the third-order Cauchy-Euler equation of the form:*

$$t^3 y'''(t) + at^2 y''(t) + bty'(t) + cy(t) = 0, \tag{14}$$

*where $a, b,$ and $c \in \mathbb{Z}$ and $t \in \mathbb{R}$. The types of solutions of* (14) *depend on the values of $a, b,$ and $c$ in the following way:*

(i) *If $a = 3m + 6, b = 3m^2 + 9m + 7$ and $c = (m+1)^3$ for some $m \in \mathbb{N} \cup \{0\}$, then there exists a distributional solution of the form:*

$$y(t) = \delta^{(m)}(t). \tag{15}$$

(ii)  If $a = 3m + n + 6, b = 3m^2 + 2mn + 9m + 3n + 7$, and $c = (m+1)^2(m+n+1)$ for some $m \in \mathbb{N} \cup \{0\}$ and $n \in \mathbb{N}$, then there exist two distributional solutions of the form:

$$y(t) = \delta^{(m)}(t) \text{ and } y(t) = \delta^{(m+n)}(t). \tag{16}$$

(iii)  If $a = 3m + 2n + 6, b = 3m^2 + 4mn + n^2 + 9m + 6n + 7$, and $c = (m+1)(m+n+1)^2$ for some $m \in \mathbb{N} \cup \{0\}$ and $n \in \mathbb{N}$, then there exist two distributional solutions of the form:

$$y(t) = \delta^{(m)}(t) \text{ and } y(t) = \delta^{(m+n)}(t). \tag{17}$$

(iv)  If $a = 2m - n + 6, b = m^2 - 2mn + 6m - 3n + 7$, and $c = (1-n)(1+m)^2$ for some $m \in \mathbb{N} \cup \{0\}$ and $n \in \mathbb{N}$, then there exist a distributional solution and a weak solution of the form:

$$y(t) = \delta^{(m)}(t) \text{ and } y(t) = H(t)\frac{t^{n-1}}{(n-1)!}. \tag{18}$$

(v)  If $a = m - 2n + 6, b = n^2 - 2nm - 6n + 3m + 7$, and $c = (m+1)(n-1)^2$ for some $m \in \mathbb{N} \cup \{0\}$ and $n \in \mathbb{N}$, then there exist a distributional solution and a weak solution of the form:

$$y(t) = \delta^{(m)}(t) \text{ and } y(t) = H(t)\frac{t^{n-1}}{(n-1)!}. \tag{19}$$

**Proof.** Applying the Laplace transform $\mathcal{L}\{y(t)\} = Y(s)$ to (14) and using Example 3(iv),(v), we obtain:

$$s^3\frac{d^3}{ds^3}Y(s) + (9-a)s^2\frac{d^2}{ds^2}Y(s) + (b - 4a + 18)s\frac{d}{ds}Y(s) + (b - 2a - c + 6)Y(s) = 0. \tag{20}$$

Suppose that a solution of (20) is of the form $Y(s) = s^r$, where $r$ is a real constant. Substituting $Y(s), Y'(s), Y''(s)$, and $Y'''(s)$ into (20), we obtain:

$$[r(r-1)(r-2) + (9-a)r(r-1) + (b - 4a + 18)r + b - 2a - c + 6]s^r = 0.$$

Since $s^r \neq 0$, we have:

$$r(r-1)(r-2) + (9-a)r(r-1) + (b - 4a + 18)r + b - 2a - c + 6 = 0,$$

or equivalently,

$$r^3 + (6-a)r^2 + (b - 3a + 11)r + b - 2a - c + 6 = 0. \tag{21}$$

Case (i): If $a = 3m + 6, b = 3m^2 + 9m + 7$, and $c = (m+1)^3$ for some $m \in \mathbb{N} \cup \{0\}$, then substituting $a, b$, and $c$ into (21) yields:

$$r^3 - 3mr^2 + 3m^2r - m^3 = 0,$$

and so:

$$(r - m)^3 = 0. \tag{22}$$

Thus, we have the repeated real roots of (22), that is $r = m$. Therefore, the solution of (20) is $Y(s) = s^m$.

Now, $Y(s)$ is an analytic function over the entire $s$-plane. Taking the inverse Laplace transform to $Y(s)$ and using Example 3(ii),(iii), we obtain a solution of (14), which is a distributional solution of the form (15).

Case (ii): If $a = 3m + n + 6, b = 3m^2 + 2mn + 9m + 3n + 7$, and $c = (m+1)^2(m+n+1)$ for some $m \in \mathbb{N} \cup \{0\}, n \in \mathbb{N}$, then substituting $a, b$ and $c$ into (21) yields:

$$r^3 - (3m + n)r^2 + (3m^2 + 2mn)r - (m^3 + m^2 n) = 0,$$

and so

$$(r - m)^2(r - (m + n)) = 0. \tag{23}$$

Thus, we have the real roots of (23), which are $r = m$ and $r = m + n$. Therefore, the solutions of (20) are $Y(s) = s^m$ and $Y(s) = s^{m+n}$.

Now, $Y(s)$ are analytic functions over the entire $s$-plane. Taking the inverse Laplace transform to $Y(s)$ and using Example 3(ii),(iii), we obtain the solutions of (14), which are distributional solutions of the form (16).

Case (iii): If $a = 3m + 2n + 6, b = 3m^2 + 4mn + n^2 + 9m + 6n + 7$, and $c = (m+1)(m+n+1)^2$ for some $m \in \mathbb{N} \cup \{0\}$ and $n \in \mathbb{N}$, then substituting $a, b$, and $c$ into (21) yields:

$$r^3 - (3m + 2n)r^2 + (3m^2 + 4mn + n^2)r - (m^3 + 2m^2 n + mn^2) = 0,$$

and so:

$$(r - m)(r - (m + n))^2 = 0. \tag{24}$$

Thus, we have the real roots of (24), which are $r = m$ and $r = m + n$. Therefore, the solutions of (20) are $Y(s) = s^m$ and $Y(s) = s^{m+n}$.

Now, $Y(s)$ are analytic functions over the entire $s$-plane. Taking the inverse Laplace transform to $Y(s)$ and using Example 3(ii),(iii), we obtain the solutions of (14), which are distributional solutions of the form (17).

Case (iv): If $a = 2m - n + 6, b = m^2 - 2mn + 6m - 3n + 7$, and $c = (1-n)(1+m)^2$ for some $m \in \mathbb{N} \cup \{0\}$ and $n \in \mathbb{N}$, then substituting $a, b$, and $c$ into (21) yields:

$$r^3 + (n - 2m)r^2 + (m^2 - 2nm)r + nm^2 = 0,$$

and so:

$$(r - m)^2(r + n) = 0. \tag{25}$$

Thus, we have the real roots of (25), which are $r = m$ and $r = -n$. Therefore, the solutions of (20) are $Y(s) = s^m$ and $Y(s) = s^{-n}$.

Clearly $Y(s)$ are analytic functions on the right half-plane $\Re(s) > 0$. Applying the inverse Laplace transform to $Y(s)$ and using Example 3(i)–(iii), we obtain the solutions of (14), which are a distributional solution and a weak solution of the form (18).

Case (v): If $a = m - 2n + 6, b = n^2 - 2nm - 6n + 3m + 7$, and $c = (m+1)(n-1)^2$ for some $m \in \mathbb{N} \cup \{0\}$ and $n \in \mathbb{N}$, then substituting $a, b$, and $c$ into (21) yields:

$$r^3 + (2n - m)r^2 + (n^2 - 2mn)r - mn^2 = 0,$$

and so:

$$(r - m)(r + n)^2 = 0. \tag{26}$$

Thus, we have the real roots of (26), which are $r = m$ and $r = -n$. Therefore, the solutions of (20) are $Y(s) = s^m$ and $Y(s) = s^{-n}$.

Now, $Y(s)$ are analytic functions over the half-plane $\Re(s) > 0$. Taking the inverse Laplace transform to $Y(s)$ and using Example 3(i)–(iii), we obtain the solutions of (14), which are a distributional solution and a weak solution of the form (19). $\quad\square$

**Theorem 2.** *The type of solutions of* (14) *whose solutions are a linear combination of distributional solutions and weak solutions depends on the values of a, b, and c in the following way:*

(i) *If $a = 3m + 2n + p + 6$, $b = 3m^2 + 9m + n^2 + n(4m + p + 6) + p(2m + 3) + 7$, and $c = (m + 1)(m + n + 1)(m + n + p + 1)$ for some $n, p \in \mathbb{N}$ and $m \in \mathbb{N} \cup \{0\}$, then all solutions are a linear combination of the distributional solutions of the form:*

$$y(t) = C_1 \delta^{(m)}(t) + C_2 \delta^{(m+n)}(t) + C_3 \delta^{(m+n+p)}(t), \tag{27}$$

*where $C_1$, $C_2$, and $C_3$ are arbitrary constants.*

(ii) *If $a = 2m + n - p + 6$, $b = m^2 + m(n - 2p + 6) + 3n - 3p - pn + 7$, and $c = (m + 1)(m + n + 1)(1 - p)$ for some $n, p \in \mathbb{N}$ and $m \in \mathbb{N} \cup \{0\}$, then all solutions are a linear combination of distributional solutions and a weak solution of the form:*

$$y(t) = C_1 \delta^{(m)}(t) + C_2 \delta^{(m+n)}(t) + C_3 H(t) \frac{t^{p-1}}{(p-1)!}, \tag{28}$$

*where $C_1$, $C_2$, and $C_3$ are arbitrary constants.*

(iii) *If $a = m - 2n - p + 6$, $b = n^2 + n(p - 2m - 6) + 3m - mp - 3p + 7$, and $c = (n - 1)(m + 1)(n + p - 1)$ for some $n, p \in \mathbb{N}$ and $m \in \mathbb{N} \cup \{0\}$, then all solutions are a linear combination of the distributional solution and weak solutions of the form:*

$$y(t) = C_1 H(t) \frac{t^{n-1}}{(n-1)!} + C_2 H(t) \frac{t^{n+p-1}}{(n+p-1)!} + C_3 \delta^{(m)}(t), \tag{29}$$

*where $C_1$, $C_2$, and $C_3$ are arbitrary constants.*

**Proof.** Applying the Laplace transform $\mathcal{L}\{y(t)\} = Y(s)$ to (14) and using Example 3(iv),(v), we obtain (20), and suppose that a solution of (20) is of the form $Y(s) = s^r$, where $r$ is a real constant. Substituting $Y(s), Y'(s), Y''(s)$ and $Y'''(s)$ into (20), we obtain (21).

Case (i): If $a = 3m + 2n + p + 6$, $b = 3m^2 + 9m + n^2 + n(4m + p + 6) + p(2m + 3) + 7$, and $c = (m + 1)(m + n + 1)(m + n + p + 1)$ for some $n, p \in \mathbb{N}$ and $m \in \mathbb{N} \cup \{0\}$, then substituting $a, b$, and $c$ into (21) yields:

$$r^3 - (3m + 2n + p)r^2 + (3m^2 + n^2 + 4mn + 2mp + np)r - (m^2 + mn)(m + n + p) = 0,$$

and so:

$$(r - m)(r - (m + n))(r - (m + n + p)) = 0. \tag{30}$$

Thus, we have the real roots of (30), which are $r = m$, $r = m + n$, and $r = m + n + p$. By the superposition principle, the solution of (20) is $Y(s) = C_1 s^m + C_2 s^{m+n} + C_3 s^{m+n+p}$, for any constants $C_1$, $C_2$, and $C_3$.

Now, $Y(s)$ is an analytic function over the entire $s$-plane. Taking the inverse Laplace transform to $Y(s)$ and using Example 3(i)–(iii), we obtain a solution of (14), which is a distributional solution of the form (27).

Case (ii): If $a = 2m + n - p + 6$, $b = m^2 + m(n - 2p + 6) + 3n - 3p - pn + 7$, and $c = (m + 1)(m + n + 1)(1 - p)$ for some $n, p \in \mathbb{N}$ and $m \in \mathbb{N} \cup \{0\}$, then substituting $a, b$, and $c$ into (21) yields:

$$r^3 + (p - 2m - n)r^2 + (m^2 + mn - 2pm - pn)r + mp(m + n) = 0,$$

and so:

$$(r - m)(r - (m + n))(r + p) = 0. \tag{31}$$

Thus, we have the real roots of (31), which are $r = m, r = m + n$, and $r = -p$. By the superposition principle, the solution of (20) is $Y(s) = C_1 s^m + C_2 s^{m+n} + C_3 s^{-p}$, for any constants $C_1$, $C_2$, and $C_3$.

Now, $Y(s)$ are analytic functions over the half-plane $\Re(s) > 0$. Taking the inverse Laplace transform to $Y(s)$ and using Example 3(i)–(iii), we obtain a solution of (14), which is a linear combination of distributional solutions and a weak solution of the form (28).

Case (iii): If $a = m - 2n - p + 6$, $b = n^2 + n(p - 2m - 6) + 3m - mp - 3p + 7$, and $c = (n - 1)(m + 1)(n + p - 1)$ for some $n, p \in \mathbb{N}$ and $m \in \mathbb{N} \cup \{0\}$, then substituting $a, b$, and $c$ into (21) yields:

$$r^3 + (2n + p - m)r^2 + (n^2 + np - 2nm - mp)r - nm(n + p) = 0,$$

and so

$$(r + n)(r + (n + p))(r - m) = 0. \tag{32}$$

Thus, we have the real roots of (32), which are $r = -n, r = -n - p$ and $r = m$. By the superposition principle, the solution of (20) is $Y(s) = C_1 s^{-n} + C_2 s^{-n-p} + C_3 s^m$, for any constants $C_1$, $C_2$, and $C_3$.

Now, $Y(s)$ are analytic functions over the half-plane $\Re(s) > 0$. Taking the inverse Laplace transform to $Y(s)$ and using Example 3(i)–(iii), we obtain a solution of (14), which is a linear combination of distributional solution and weak solutions of the form (29).　□

**Theorem 3.** *The distributional solutions of* (14) *depend on the values of a, b, and c of the form:*

$$h^3 + (6 - a)h^2 + (b - 3a + 11)h + b - 2a - c + 6 = 0, \tag{33}$$

*where $h \in \mathbb{N} \cup \{0\}$ is the order of the distributional solutions.*

**Proof.** By Lemma 2, substituting $n = 3, a_0(0) = c, a_1(0) = b, a_2(0) = a$, and $a_3(0) = 1$ into (10) and using (33) yield:

$$\sum_{i=0}^{3} (-1)^i a_i(0)(h + i)! = ch! + (-1)b(h + 1)! + (-1)^2 a(h + 2)! + (-1)^3(1)(h + 3)!$$

$$= h! \left[ c - b(h + 1) + a(h + 2)(h + 1) - (h + 3)(h + 2)(h + 1) \right]$$

$$= h! \left[ c - (h + 1)(b + h^2 + 5h + 6 - ah - 2a) \right]$$

$$= h! \left[ c - (bh + h^3 + 5h^2 + 6h - ah^2 - 2ah + b + h^2 + 5h + 6 - ah - 2a) \right]$$

$$= h! \left[ c - (h^3 + (6 - a)h^2 + (b - 3a + 11)h + (b - 2a + 6)) \right]$$

$$= 0.$$

This completes the proof.　□

**Remark 1.** *The value of $r$ in* (21) *and the value of $h$ in* (33) *are identical.*

**Example 4.** *From Theorem 1(i), if $m = 1$, then* (14) *becomes:*

$$t^3 y'''(t) + 9t^2 y''(t) + 19t y'(t) + 8y(t) = 0. \tag{34}$$

*It follows from* (15) *that its distributional solution is:*

$$y(t) = \delta'(t). \tag{35}$$

*From Theorem 1(ii), if m and n = 1, then (14) becomes:*

$$t^3 y'''(t) + 10t^2 y''(t) + 24ty'(t) + 12y(t) = 0. \tag{36}$$

*It follows from (16) that its distributional solutions are:*

$$y(t) = \delta'(t) \ and \ y(t) = \delta''(t). \tag{37}$$

*From Theorem 1(iii), if m and n = 1, then (14) becomes:*

$$t^3 y'''(t) + 11t^2 y''(t) + 30ty'(t) + 18y(t) = 0. \tag{38}$$

*It follows from (17) that its distributional solutions are:*

$$y(t) = \delta'(t) \ and \ y(t) = \delta''(t). \tag{39}$$

*From Theorem 1(iv), if m and n = 2, then (14) becomes:*

$$t^3 y'''(t) + 8t^2 y''(t) + 9ty'(t) - 9y(t) = 0. \tag{40}$$

*It follows from (18) that its distributional solution and weak solution are:*

$$y(t) = \delta''(t) \ and \ y(t) = H(t)t. \tag{41}$$

*Moreover, from Theorem 1(v), if m and n = 3, then (14) becomes:*

$$t^3 y'''(t) + 3t^2 y''(t) - 11ty'(t) + 16y(t) = 0. \tag{42}$$

*It follows from (19) that its distributional solution and weak solution are:*

$$y(t) = \delta^{(3)}(t) \ and \ y(t) = H(t)\frac{t^2}{2!}. \tag{43}$$

*By applying (13), it is easy to verify that (35), (37), and (39) satisfy (34), (36), and (38), respectively. By applying (12) and (13), it is easy to verify that (41) and (43) satisfy (40) and (42), respectively.*

**Example 5.** *From Theorem 2(i), if m, n, and p = 1, then (14) becomes:*

$$t^3 y'''(t) + 12t^2 y''(t) + 36ty'(t) + 24y(t) = 0. \tag{44}$$

*It follows from (27) that its distributional solution is:*

$$y(t) = C_1 \delta'(t) + C_2 \delta''(t) + C_3 \delta'''(t). \tag{45}$$

*From Theorem 2(ii), if m, n, and p = 2, then (14) becomes:*

$$t^3 y'''(t) + 10t^2 y''(t) + 15ty'(t) - 15y(t) = 0. \tag{46}$$

*It follows from (28) that its solution contains both distributional solutions and a weak solution, namely,*

$$y(t) = C_1 \delta''(t) + C_2 \delta^{(4)}(t) + C_3 H(t)t. \tag{47}$$

*Moreover, from Theorem 2(iii), if m = n = 3 and p = 1, then (14) becomes:*

$$t^3 y'''(t) + 2t^2 y''(t) - 14ty'(t) + 24y(t) = 0. \tag{48}$$

*It follows from* (29) *that its solution contains both a distributional solution and weak solutions, namely,*

$$y(t) = C_1 \delta^{(3)}(t) + C_2 H(t)\frac{t^2}{2!} + C_3 H(t)\frac{t^3}{3!}. \tag{49}$$

*By applying* (13), *it is easy to verify that* (45) *satisfies* (44). *By applying* (12) *and* (13), *it is easy to verify that* (47) *and* (49) *satisfy* (46) *and* (48), *respectively.*

## 4. Conclusions

We used the Laplace transform technique to find the generalized solutions of the third-order Cauchy-Euler equation of the form:

$$t^3 y'''(t) + a t^2 y''(t) + b t y'(t) + c y(t) = 0,$$

where $a$, $b$, and $c \in \mathbb{Z}$ and $t \in \mathbb{R}$. Then, we took the inverse Laplace transform to the derived solutions. We found the conditions of $a$, $b$, and $c$, for the case of one distributional solution, two distributional solutions, three distributional solutions, containing both a distributional solution and a weak solution, a linear combination of two distributional solutions and one weak solution, and a linear combination of one distributional solution and two weak solutions. It should be noted here that the inverse Laplace transform of $s^n$, where $n \in \mathbb{N}$ as shown in Examples 4 and 5, however, for the classical solutions is not mentioned here, which can be found in Greenberg [8].

**Author Contributions:** The order of the author list reflects contributions to the paper.

**Funding:** This research received no external funding.

**Acknowledgments:** The second author was financially supported by the National Research Council of Thailand and Faculty of Science, Khon Kaen University 2019.

**Conflicts of Interest:** The authors declare no conflict of interest.

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
