# Peer review of "Generalized Solutions of the Third-Order Cauchy-Euler Equation in the Space of Right-Sided Distributions via Laplace Transform"

_mathematics, doi:10.3390/math7040376_

Round 1
Reviewer 1 Report
This manuscript deals with 'generalized solution of Euler-Cauchy equation in the space of right-sided distributions via Laplace transform, and this manuscript has a originality and there is a value to publish. Moreover, since this manuscript shows the role of a guide in the study of Euler-Cauchy equation, I would like to recommend as 'Accept in present form'.
* There is a typo.
Line 37 of P2; by Kim. [23]. ===> by Kim {23]. (delete period)
Line 42 of P2; I think it is better to rewrite this sentence
(whether split into two sentences, or delete 'contains')
Thanks, and hope you have a good day.
Author Response
I would like to thank the referee for generous advice and for valuable suggestions to improved our manuscript. I have changed the sentences according to the suggestion by reviewer comments.
Reviewer 2 Report
Dear authors, I was glad to read your manuscript. Your article fits well for Mathematics journal and can be published after revision. Please, find my suggestions below.
I found the typo in line 39. You are referred to the article in which the parameter "m" is mentioned. However, in your formula, this parameter is named as "c".
Please intensively check the spelling, grammar, and style of the manuscript. I noticed some errors in lines 38, 39, 41 and 173 for example. Word repeating in the sentences in lines 25 and 32 also can be fixed.
Please expand the abstract section and show contribution over your previous papers in the introduction.
Author Response

(The authors gave the same response as above.)
